

# Mass media pressure on physical build, psychological well-being and physical-healthy profile. An explanatory model in adulthood

Eduardo Melguizo-Ibáñez[1], Gabriel González-Valero[1],
Georgian Badicu[2], Wilhelm Robert Grosz[2], Marius Bazgan[3] and
Pilar Puertas Molero[1]

[1] Department of Didactics of Musical, Plastic and Corporal Expression, University of Granada, Granada, Spain
[2] Department of Physical Education and Special Motricity, Transilvania University of Brasov, Romania, BRASOV, România
[3] Department of Psychology, Education and Teacher Training, Transilvania University of Brasov, BRASOV, România

Corresponding author
Georgian Badicu,
georgian.badicu@unitbv.ro

## ABSTRACT

**Background:** The pressure exerted by the media on mental image, psychological well-being and the physical-dietary sphere is of vital interest in understanding human behavioral patterns at different stages of development. The present research reflects the objectives of developing an explanatory model of the associations between media pressure and physical build on psychological well-being, physical activity and the Mediterranean diet and testing the structural model using a multi-group model according to participants' stage of adulthood development.

**Methods:** A descriptive, non-experimental, cross-sectional study was carried out on a sample of 634 participants aged between 18 and 65 years (35.18 ± 9.68). An *ad hoc* socio-demographic questionnaire, Sociocultural Attitudes Towards Appearance Questtionnaire-4, Psychological Well-Being Scale, Predimed questtionnaire and International Physical Activity Questionnaire-Short Form were used for data collection.

**Results:** Participants in early adulthood show higher scores for media pressure and for pressure on physical build. It is also observed that participants in middle adulthood show higher scores for psychological well-being and physical activity levels.

**Conclusions:** In conclusion, it can be seen that there are a large number of physical, physical-health and psychological differences in each of the phases of adulthood.

## INTRODUCTION

Currently, the branch of study related to developmental psychology focuses on broadening and deepening knowledge about the different stages of the life cycle of individuals (*Blank, 1979*) as well as the social and cultural changes that occur. Adulthood is characterised by

the fact that the person is able to live independently without the need for emotional, social, affective and economic guardianship by parents (*Ribeiro, Yassuda & Neri, 2020*). During this stage of development, biological maturity is reached, which refers to the full development of physical capacities (*Vleioras, 2021*); however, psychological maturity is also reached, with the full development of affective and intellectual functions (*Tanner, Arnett & Leis, 2008*). This stage is preceded by adolescence, which is characterised by major physical, emotional, psychological and dietary changes (*Vernetta-Santana et al., 2018*). *Byrne (2005)*, states that adolescence is not a continuous process, but is composed of three phases: Early adolescence (11–14 years), middle adolescence (15–17 years) and late adolescence (18–21 years).

According to *Levinson (1986)* and *Papalia et al. (2009)*, the adult stage is made up of three stages: early adulthood (18–45 years), middle adulthood (40–60 years) and late adulthood (from 60 years), the first stage coinciding with the end of adolescence. In view of this finding, *Becht et al. (2021)* state that the onset of adulthood is variable and dependent on the characteristics of each individual, with some people starting earlier. During early adulthood the subject's life is characterised by increased dynamism and vitality (*Berger & Thompson, 2001*), during middle adulthood subjects participate fully in different social activities, strengthening their professional life and the roles they play in society (*Becht et al., 2021*) and mature adulthood is characterised by a decline in physiological, psychological and sporting aspects (*Becht et al., 2021*).

In order to understand different social behaviours, it is necessary to understand and highlight the importance of the media, as well as the influence of the media on the acquisition of certain behaviours (*Rodgers, Sales & Chabrol, 2010*). Studies by *Rodgers, Sales & Chabrol (2010)* and *Puertas-Molero, Marfil-Carmona & González-Valero (2021)* claim that the media can negatively affect psychological well-being, specifically by exerting pressure on individual body image through the transmission of different standards of beauty. Despite the above, such technologies have been shown to have a positive effect on different psychosocial aspects, psychological and emotional well-being (*Puertas-Molero, Marfil-Carmona & González-Valero, 2021*). It has also been shown that media can be used as a tool to support the acquisition of healthy and physically active lifestyles (*Marfil-Carmona et al., 2021*).

A healthy and active lifestyle has numerous benefits for people's physical and mental health (*Palma-Leal et al., 2021*; *Chillón et al., 2021*). Physical activity can be defined as any skeletal movement produced by muscle contraction that results in significant energy expenditure (*Escalante, 2011*). In this case, the regular practice of physical activity produces numerous benefits such as the reduction of hypertension and hypercholesterolemia, as well as preventing the onset of various types of cancer, among other diseases (*Pojednic et al., 2022*). Likewise, *Sanz-Martín et al. (2022)* state that positive repercussions are also found at a social and psychological level. In this case, for the adult population, regular physical exercise affects quality of life in three psychological aspects: distraction, self-efficacy and socialisation (*Melguizo-Ibáñez et al., 2021*). In addition, physical exercise acts as an effective non-pharmacological therapy against the main diseases associated with

population ageing (*Aparicio García-Molina, Carbonell-Baeza & Delgado-Fernández, 2010*), slowing down the deterioration of nerve cells (*Herrero & Ferradaz, 2011*).

At the same time, positive adherence to a healthy dietary pattern is one of the most influential elements on people's health (*Zurita-Ortega et al., 2018*). The traditional Mediterranean diet is characterised by a healthy pattern rich in plant foods, vegetable fats such as olive oil together with a high or medium intake of fish and a reduced consumption of red meat (*Muros et al., 2017*). Currently, the adult population does not have a good adherence to this dietary pattern due to an increased intake of high-calorie ready meals (*Melguizo-Ibáñez et al., 2022a*), resulting in an increase in the number of obese people in Western societies (*Melguizo-Ibáñez et al., 2020*).

Therefore, on the basis of the above, the present research reflects the following research hypotheses:

**H.1.** Participants in early adulthood will show higher levels of physical activity.

**H.2.** Subjects in early adulthood will show greater media pressure, as well as greater pressure for body build than those in middle or late adulthood.

**H.3.** Participants in middle or late adulthood will show better scores on psychological well-being and adherence to the Mediterranean diet than subjects in early adulthood.

The research objectives presented are: To identify and establish the relationships between media pressure and physical build on psychological well-being, physical activity and the Mediterranean diet. This is broken down into (a) developing an explanatory model of the associations between media pressure and physical build on psychological well-being, physical activity and the Mediterranean diet and (b) testing the structural model using a multi-group model according to the developmental stage of adulthood of the participants.

## MATERIALS AND METHODS

### Design and participants

A non-experimental (*ex post facto*), descriptive, cross-sectional research was carried out. A single data collection was carried out with a single group using convenience sampling, with the final sample comprising a total of 634 participants. In this case, the only criterion for inclusion was the possession of a university degree in teaching with a specialisation in physical education. This meant that some participants were still in the process of obtaining this degree and others were in the profession. Regarding gender distribution of the participants, 55.5% ($n = 352$) were male and 44.5% were female ($n = 282$), with the age of the participants ranging from 19 to 66 years ($35.18 \pm 9.68$). To ensure that the questions were not answered randomly, two questions were duplicated, eliminating participants who did not answer these two questions in the same way. In total 48 questionnaires were eliminated. In this research it has been received the written informed consent from participants of our study.

## Variables and instruments

The instruments used are described below:

*Ad hoc socio-demographic questionnaire* for the collection of demographic variables such as gender and age. In order to classify the different stages of development, the recommendations established by *Papalia et al. (2009)* and *Levinson (1986)* have been followed: Early adulthood (18–40 years), middle adulthood (41–60 years) and late adulthood (60 years and older).

*Sociocultural Attitudes Towards Appearance Questtionnaire-4 (SATAQ-4)* (*Schaefer et al., 2015*). For this research it has been used the Spanish versión carried out by *Llorente et al. (2014)*. This instruments measures data related to media pressure (items 19, 20, 21 and 22) and personal physical appearance (items 1, 2, 6, 7 and 10). The reliability of the questionnaire obtained a value of $\alpha = 0.918$, however, for media pressure a value of $\alpha = 0.897$ was obtained and for personal physical appearance $\alpha = 0.905$.

*Psychological Well-Being Scale (PWBS)* (*Ryff & Keyes, 1995*): For this study it has been used the Spanish version adapted by *Díaz et al. (2006)*. The scale consists of a Likert scale with six response options ("1 = strongly disagree" to "6 = strongly agree"). It provides a summary of psychological well-being that emerges from the following six dimensions of psychological well-being: self-acceptance (items 1, 7, 19 and 31), positive relationships (items 2, 8, 14, 26 and 32), autonomy (items 3, 4, 9, 15, 21 and 27), mastery of the environment (items 5, 11, 16, 22 and 39), personal growth (items 24, 36, 37 and 38) and purpose in life (items 6, 12, 17, 18 and 23). In this case, Cronbach's alpha obtained a score of $\alpha = 0.964$.

*Predimed questtionnaire* (*Schröder et al., 2011*): This questionnaire was used to obtain the Mediterranean diet variable. This instrument is composed of 14 items, where once all of them are answered, a final score is obtained that categorises the participants' responses into three levels: low adherence ($\leq 7$), medium adherence (8–10) and high adherence ($>10$). In this case, Cronbach's alpha obtained a score of $\alpha = 0.810$.

*International Physical Activity Questionnaire-Short Form (IPAQ-SF)*, *Booth (2000)*: This instrument is a self-report instrument to record the intensity of physical activity (light, moderate and vigorous activity). It consists of a total of seven items measuring the frequency (days per week) and duration (time per day) of different physical activities. In terms of the distribution of the items, two are related to vigorous activities, two to moderate activities, two to walking and finally one item to measure the degree of physical inactivity.

## Procedure

This research followed a data collection procedure similar to that proposed by *Melguizo-Ibáñez et al. (2022a)*. To carry out this study, the first step was to conduct a literature review on the subject to understand the issues involved. A google form was then created with the data collection instruments, research objectives and informed consent of the participants. Due to the pandemic caused by the COVID-19 virus, data were collected telematically. This method of data collection was chosen due to mobility restrictions imposed by the Spanish government. In order to avoid randomisation of responses, two
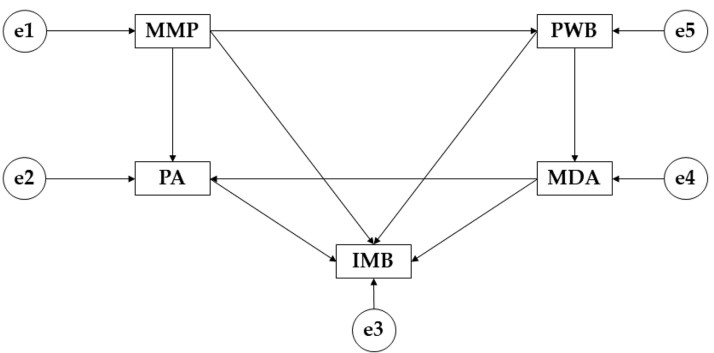

**Figure 1 Proposed theoretical model.** Mediterranean Diet Adherence (MDA); Mass Media Pressure (MMP); Psychological Well Being (PWB); Internalising Muscular Build (IMB); Physical Activity (PA).

questions were duplicated, eliminating those questionnaires where they did not coincide. On the basis of the above, 48 questionnaires were eliminated. Finally, the study complied with the ethical criteria established in the Declaration of Helsinki and was approved and supervised by an ethics committee of the University of Granada (2966/CEIH/2022).

## Data analysis

Statistical analysis of the data was carried out with the IBM SPSS Statics 25.0 programme (IBM Corp, Armonk, NY, USA). Firstly, an analysis of the normality of the sample was carried out using the Kolmogorov-Smirnov test, obtaining a normal nomality. Next, a single-factor ANOVA was performed, using Pearson's chi-square test to study statistically significant differences. In this case, the significance level was set at $p < 0.05$. To study the magnitude of the effect, Cohen's standardised d (*Cohen, 1992*) was used. These values have been interpreted as follows: null (0.0–0.19), small (0.20–0.49), medium (0.50–0.79) and large (≥0.80). Furthermore, this data analysis is based on the study carried out by *Melguizo-Ibáñez et al. (2022b)*.

The IBM SPSS Amos 26.0 software (IBM Corp, Armonk, NY, USA) was used to create the structural equation models, which allows us to establish the relationships between the variables that make up this theoretical model (Fig. 1). Three models have been proposed according to the different phases of adulthood. For endogenous variables, causal explanations were made taking into account the observed associations between indicators and measurement reliability, therefore, measurement error of observable variables was included in the model and could be directly controlled and interpreted as multivariate regression coefficients. Furthermore, one-way arrows represented lines of influence between the latent variables and were interpreted from the regression weights. The data analysis is based on the study carried out by *Melguizo-Ibáñez et al. (2022b)*.

To evaluate the model, the recommendations of *Bentler (1990)*, *McDonald & Marsh (1990)*, *Maydeu-Olivares (2017)* and *Kyriazos (2018)* have been followed. To assess the goodness of fit, chi-square should be used, where non-significant *p*-values indicate a good fit of the model. Likewise, for the Comparative Fit Index (CFI), Goodness of Fit Index (GFI) and Incremental Reliability Index (IFI) values above 0.900 indicate a good model fit,

**Table 1 Comparative study of variables.**

|  |  | N | M | S.D | F | p | ES (d) | 95% CI |
|---|---|---|---|---|---|---|---|---|
| MDA | Early adulthood | 428 | 7.50 | 2.61 | 8.131 | ≤0.05 | 0.546[a] | [0.256–0.836][a] |
|  | Middle adulthood | 154 | 7.89 | 2.00 |  |  | 0.513[b] | [0.194–0.831][b] |
|  | Older adulthood | 52 | 8.88 | 1.71 |  |  |  |  |
| MMP | Early adulthood | 428 | 2.33 | 0.72 | 3.490 | ≤0.05 | 0.364[a] | [0.075–0.653][a] |
|  | Middle adulthood | 154 | 2.24 | 0.73 |  |  |  |  |
|  | Older adulthood | 52 | 2.07 | 0.66 |  |  |  |  |
| PWB | Early adulthood | 428 | 4.54 | 0.71 | 3.055 | ≤0.05 | 0.282[c] | [−0.286–0.290][c] |
|  | Middle adulthood | 154 | 4.70 | 0.65 |  |  |  |  |
|  | Older adulthood | 52 | 4.58 | 0.72 |  |  |  |  |
| IMB | Early adulthood | 428 | 2.82 | 1.07 | 1.702 | 0.183 | NP | NP |
|  | Middle adulthood | 154 | 2.67 | 1.06 |  |  |  |  |
|  | Older adulthood | 52 | 2.61 | 0.85 |  |  |  |  |
| PA | Early adulthood | 428 | 1.26 | 0.44 | 1.786 | 0.168 | NP | NP |
|  | Middle adulthood | 154 | 1.29 | 0.45 |  |  |  |  |
|  | Older adulthood | 52 | 1.15 | 0.36 |  |  |  |  |

**Notes:**
[a] Differences between Early and Mature Adulthood.
[b] Differences between Middle and Mature Adulthood.
[c] Differences between Middle and Early Adulthood.
Mediterranean Diet Adherence (ADM); Media Pressure (PMM); Psychological Well-Being (BPS); Internalised Muscle Building (IMB); Physical Activity (PA). Mediterranean Diet Adherence (MDA); Media Pressure (MMP); Psychological Well-Being (PWB); Internalised Muscular Building (IMB); Physical Activity (PA).

while for the Root Mean Square Approximation (RMSEA) values below 0.100 indicate a good model fit.

# RESULTS

Based on the results obtained in the comparative analysis (Table 1), it is observed that participants in mature adulthood (M = 8.88) show better adherence to the Mediterranean diet than those in the intermediate phase (M = 7.89) or those in the early phase (M = 7.50). Regarding media pressure, it is observed that adults in the early stage (M = 2.33) show higher scores than those in the middle (M = 2.24) or late stage (M = 2.07). It is also observed that the pressure exerted with muscular build is higher for adults in the early phase (M = 2.82) than for adults in the middle (M = 2.67) and mature phase (M = 2.61). In relation to psychological well-being, it is observed that adults in the middle phase (M = 4.70) show better scores than those in the mature phase (M = 4.58) or early phase (M = 4.54). Finally, for the practice of physical activity, it is observed that average adults (M = 1.29), show a higher weekly physical exercise time than adults in the early (M = 1.26) or mature phase (M = 1.15).

Continuing with the data from the structural equation models, the one proposed for early adulthood showed a good fit for each of the component indices. The chi-square analysis obtained a significant p-value ($X^2$ = 10.139; df = 2; pl = 0.006). These data cannot be interpreted independently due to the influence of sample size and susceptibility (*Tenenbaum & Eklund, 2007*); other standardised indices that were less sensitive to sample
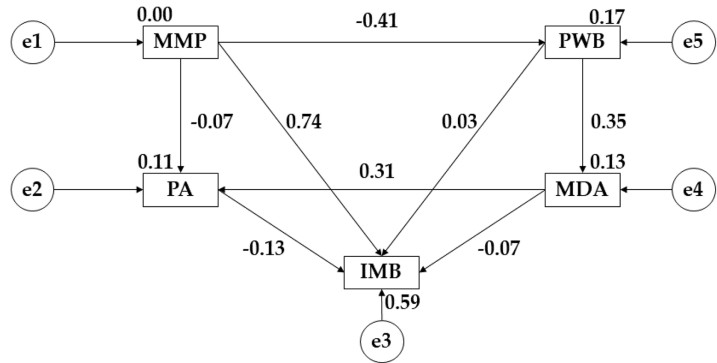

**Figure 2 Theoretical model proposed and developed for early adulthood.** Mediterranean Diet Adherence (MDA); Mass Media Pressure (MMP); Psychological Well Being (PWB); Internalising Muscular Build (IMB); Physical Activity (PA).

**Table 2 Structural model of the theoretical model developed for early adulthood.**

| Associations between variables | R.W. | | | | S.R.W. |
|---|---|---|---|---|---|
| | Stimations | S.E. | C.R. | *p* | Stimations |
| PWB ← MMP | −0.406 | 0.044 | −9.265 | *** | −0.409 |
| MDA ← PWB | 1.299 | 0.166 | 7.836 | *** | 0.355 |
| PA ← MMP | −15.601 | 10.815 | −1.443 | 0.149 | −0.067 |
| PA ← MDA | 20.147 | 2.975 | 6.772 | *** | 0.313 |
| IMB ← MMP | 1.106 | 0.051 | 21.713 | *** | 0.740 |
| IMB ← PWB | 0.049 | 0.054 | 0.907 | 0.364 | 0.033 |
| IMB ← MDA | −0.029 | 0.014 | −2.008 | ** | −0.070 |
| IMB ← PA | −0.001 | 0.003 | −3.939 | *** | −0.129 |

**Notes:**
** $p \leq 0.05$.
*** $p \leq 0.001$.
Mediterranean Diet Adherence (MDA); Mass Media Pressure (MMP); Psychological Well Being (PWB); Internalising Muscular Build (IMB); Physical Activity (PA).

size have therefore been used. The CFI and IFI obtained a value of 0.986, the TLI evidenced a value of 0.928, the RFI showed a value of 0.912 and finally the NFI presented a value of 0.982. The RMSEA obtained a value of 0.049.

Figure 2 and Table 2 show the results obtained for early adulthood. In this case, a negative relationship is observed for media pressure (MMP) with psychological well-being (PWB) (r = −0.409; $p \leq 0.001$) and physical activity (PA) (r = −0.067); however, a positive relationship is obtained with pressure on muscular build (IMB) (r = 0.740; $p \leq 0.001$). Continuing with psychological well-being (PWB), a positive relationship was observed with adherence to the Mediterranean diet (r = 0.355; $p \leq 0.001$) and with the pressure on muscular complexion (IMB) (r = 0.033). Adherence to the Mediterranean diet (MDA) was positively related to physical activity (PA) (r = 0.313; $p \leq 0.001$) and negatively related to muscular build (IMB) (r = 0.070; $p < 0.05$). Finally, physical activity (PA) showed a negative relationship with the pressure on muscle mass (IMB) (r = −0.129; $p \leq 0.001$).

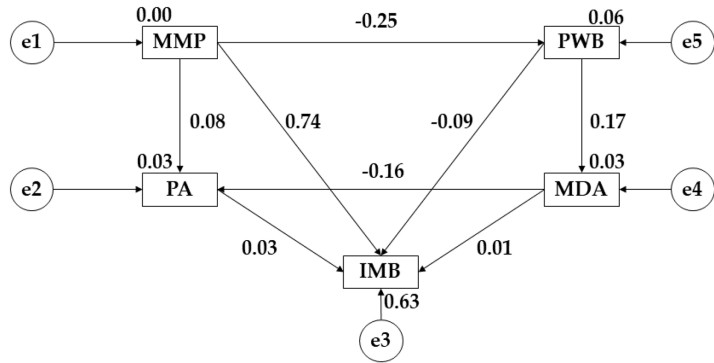

**Figure 3 Theoretical model proposed and developed for middle adulthood.** Mediterranean Diet Adherence (MDA); Mass Media Pressure (MMP); Psychological Well Being (PWB); Internalising Muscular Build (IMB); Physical Activity (PA).  

**Table 3 Structural model of the theoretical model developed for middle adulthood.**

| Associations between variables | R.W. | | | | S.R.W. |
|---|---|---|---|---|---|
| | Stimations | S.E. | C.R. | $p$ | Stimations |
| PWB ← MMP | −0.219 | 0.070 | −3.138 | *** | −0.245 |
| MDA ← PWB | 0.528 | 0.246 | 2.151 | ** | 0.171 |
| PA ← MMP | 0.047 | 0.049 | 0.962 | 0.336 | 0.076 |
| PA ← MDA | −0.036 | 0.018 | −2.028 | ** | −0.161 |
| IMB ← PA | 0.061 | 0.119 | 0.518 | 0.605 | 0.026 |
| IMB ← MDA | 0.002 | 0.027 | 0.079 | 0.937 | 0.004 |
| IMB ← PWB | −0.156 | 0.085 | −1.847 | 0.065 | −0.095 |
| IMB ← MMP | 1.120 | 0.075 | 14.978 | *** | 0.761 |

**Notes:**
** $p \leq 0.05$.
*** $p \leq 0.001$.
Mediterranean Diet Adherence (MDA); Mass Media Pressure (MMP); Psychological Well Being (PWB); Internalising Muscular Build (IMB); Physical Activity (PA).

The model developed for middle adulthood showed adequate values for each of its component indices, with the chi-square analysis showing a non-significant $p$-value ($X^2 = 21.593$; df = 2; pl = 0.000). The CFI obtained a value of 0.903, the IFI showed a score of 0.918, the TLI obtained a value of 0.938, the RFI was 0.920 and the NFI reflected a score of 0.956. For this model the RMSEA was 0.093.

Figure 3 and Table 3 show the relationships for participants in middle adulthood. In this case, for media pressure (MMP) a negative relationship is observed with psychological well-being (PWB) ($r = −0.245$; $p \leq 0.001$), however, positive relationships were found with physical activity (PA) ($r = 0.076$), and with pressure on muscular build (IMB) ($r = 0.761$; $p \leq 0.001$). For psychological well-being (PWB), a positive relationship was observed with adherence to the Mediterranean diet (MDA) ($r = 0.171$; $p \leq 0.05$), but a negative link was found with muscular build (IMB) ($r = −0.095$). For adherence to a healthy dietary pattern (MDA), there is a negative association with physical activity (PA) ($r = −0.161$; $p \leq 0.05$)
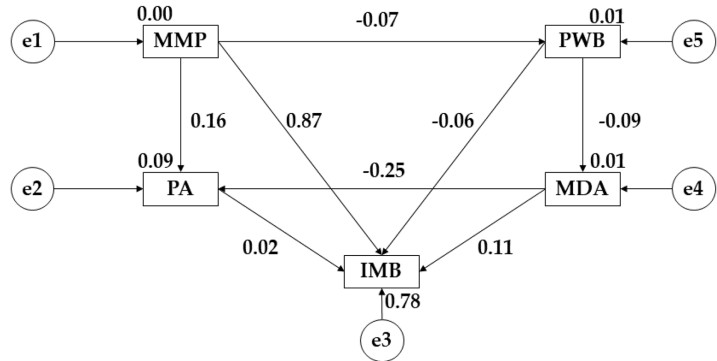

**Figure 4 Theoretical model proposed and developed for older adulthood.** Mediterranean Diet Adherence (MDA); Mass Media Pressure (MMP); Psychological Well Being (PWB); Internalising Muscular Build (IMB); Physical Activity (PA).               

**Table 4 Structural model of the theoretical model developed for late adulthood.**

| Associations between variables | R.W. | | | | S.R.W. |
|---|---|---|---|---|---|
| | Stimations | S.E. | C.R. | p | Stimations |
| PWB ← MMP | −0.081 | 0.152 | −0.530 | 0.596 | −0.074 |
| MDA ← PWB | −0.218 | 0.328 | −0.664 | 0.507 | 0.093 |
| PA ← MDA | −0.052 | 0.029 | −1.832 | 0.067 | −0.245 |
| PA ← MMP | 0.090 | 0.074 | 1.224 | 0.221 | 0.164 |
| IMB ← PWB | −0.072 | 0.076 | −0.937 | 0.349 | −0.061 |
| IMB ← MMP | 1.103 | 0.084 | 13.095 | *** | 0.867 |
| IMB ← PA | 0.053 | 0.157 | 0.334 | 0.739 | 0.023 |
| IMB ← MDA | 0.056 | 0.033 | 1.670 | 0.095 | 0.113 |

**Notes:**
*** $p \leq 0.001$.
Mediterranean Diet Adherence (MDA); Mass Media Pressure (MMP); Psychological Well Being (PWB); Internalising Muscular Build (IMB); Physical Activity (PA).

and a positive association with muscular build (IMB) (r = 0.004). For physical activity (PA) there is a positive relationship with muscular build (IMB) (r = 0.026).

The model developed for late adulthood showed adequate values for each of its component indices, with the chi-square analysis showing a non-significant $p$-value ($X^2$ = 2.707; df = 2; pl = 0.169). The CFI obtained a value of 0.991, the IFI showed a score of 0.992, the TLI obtained a value of 0.954, the RFI was 0.944 and the NFI reflected a score of 0.979. For this model the RMSEA was 0.019.

Figure 4 and Table 4 show the relationships for participants in late adulthood. For media pressure (MMP), there is a negative relationship with psychological well-being (r = −0.074), with a positive relationship with pressure on muscular build (IMB) (r = 0.867; $p \leq 0.001$) and physical activity (r = 0.164). Continuing with psychological well-being (PWB), a positive relationship was observed with adherence to the Mediterranean diet (r = 0.093) and a negative link with the pressure on muscular complexion (IMB) (r = −0.061). Focusing attention on adherence to a healthy dietary pattern (MDA), a negative relationship is observed with physical activity (r = −0.245) and a positive

relationship with the pressure on muscle mass (IMB) (r = 0.113). Finally, for physical activity (PA), a positive relationship is observed with the pressure on muscular complexion (IMB) (r = 0.023).

## DISCUSSION

The present research shows the relationships between media pressure and physical build on adherence to the Mediterranean diet, physical activity and psychological well-being at different stages of adulthood. The results obtained respond to the objectives initially set out, therefore, the present discussion aims to compare the results obtained with other research carried out previously.

Comparative analysis shows that participants in early adulthood show higher scores for media pressure and for pressure on physical build. Research by *Hong & Kim (2019)* states that people during adolescence are highly influenced by external agents found in different media or social networks. Likewise, both social networks and the media transmit body ideals, which young people want to replicate in order to gain greater social recognition (*Burnette, Kwitowski & Mazzeo, 2017*).

It was also observed that participants in middle adulthood show better scores on psychological well-being and levels of physical activity. *Lawford, Ramey & Hood (2021)* state that higher levels of physical exercise show improvements in psychological well-being, positively impacting on people's mental health. Similarly, *Murray et al. (2021)* state that a number of healthy patterns and behaviours are acquired during adolescence that tend to be maintained into adulthood. Failure to create such a habit during adolescence may result in a failure to maintain this pattern during adulthood and thus worsen the health of individuals (*Li et al., 2021*; *Yang et al., 2022*).

For subjects in mature adulthood, it was observed that this group is the one with the best adherence to the Mediterranean diet. In view of these findings, *Meng et al. (2021)* state that during early adulthood there is a detachment from adherence to a healthy dietary pattern, as there is an increased intake of high-calorie dishes. Furthermore, *Orio et al. (2016)* state that entering the labour market limits the time available to prepare healthy dishes, opting instead to eat precooked meals.

Continuing with structural equation modelling, negative relationships were observed between psychological well-being and media pressure. In view of these results, *Marfil-Carmona et al. (2021)* and *González-Valero et al. (2022)* argue that the media can play a negative role on psychological well-being. In this case, during COVID-19 confinement, it was found that continuous exposure to news related to the pandemic had a negative impact on people's mental health (*Eden et al., 2020*). On the contrary, a positive relationship was found between adherence to the Mediterranean diet and psychological well-being, stating *Muros et al. (2017)* that a healthy diet has a positive impact on people's mental image, helping people's psychological well-being.

There is also a negative relationship between physical activity and adherence to a healthy dietary pattern. Distant results were obtained by *Sbert et al. (2022)* stating that when an active lifestyle is pursued, healthy eating tends to be pursued in tandem. Similarly, *Sabingoz & Dogan (2019)* and *San Román-Mata (2019)* state that nutrition is currently

worsening due to a lack of educational programmes aimed at nutrition education. Considering the relationship between media pressure and physical activity, a negative relationship is observed in early adulthood, with a negative link in middle and late adulthood. In view of this finding, *Levinger & Hill (2020)* affirm that the media have a positive relationship with active lifestyles, since during the COVID-19 pandemic, the media were used for physical exercise.

Focusing attention on the pressure on physical build, negative relationships between this variable and adherence to the Mediterranean diet together with physical activity are observed for participants in early adulthood, while the opposite is true for those in middle or late adulthood. In view of these findings, *Pastor, Balaguer & García-Merita (2006)* state that an active and healthy lifestyle has a positive impact on a person's fitness, so that the pressure received from the environment on physical fitness is reduced. Furthermore, *Vieira et al. (2015)* state that this type of pressure tends to occur mostly in adolescents and not in adults.

Finally, the relationship between psychological wellbeing and physical build pressure turns out to be positive in early adulthood and negative in middle and late adulthood. The study conducted by *Krogh (2022)* found that when people are dissatisfied with their physical state, they tend to have a negative self-concept, which has a negative impact on people's mental health. Similarly, *Canali et al. (2021)* state that special care should be taken in young people as disorders associated with mental image, as well as eating disorders such as bulimia, may occur.

## Limitations and future perspectives

In this case, the present research reflects a series of limitations which are outlined below. The first limitation is that as this is a cross-sectional study, it is not possible to establish the cause-effect relationships of the variables over a longitudinal period of time, but only at that point in time. Furthermore, questionnaires were used as a measuring instrument, which have an intrinsic measurement error. It should be noted that the participants belonged to a very specific geographical area, so that it is not possible to generalise to a wider area of the national or regional geography.

With a view to future perspectives, an intervention programme is being carried out in which the subjects accept themselves as they are and know how to channel the pressure exerted by the media on their body image and on an active and healthy lifestyle.

## CONCLUSIONS

For the comparative analysis it is observed that participants in early adulthood are those who show higher levels of media pressure and physical build. Similarly, subjects in middle adulthood show higher levels of physical activity and psychological well-being. Likewise, participants in middle adulthood are those who show the best adherence to the Mediterranean diet.

Focusing on structural equation modelling, a negative relationship is observed between media pressure and psychological well-being. In contrast, a positive relationship is observed between media pressure and pressure on physical build. At the same time, a

positive relationship between media pressure and physical activity is observed for middle and late adulthood, while this link is negative for early adulthood.

Regarding psychological well-being, a positive relationship is observed with adherence to a healthy dietary pattern. A positive relationship is also obtained between psychological well-being and pressure on physical build for early adulthood, with a negative relationship for middle and late adulthood.

Adherence to the Mediterranean diet shows a positive relationship with physical activity only in early adulthood. At the same time, a positive relationship is found for adherence to the Mediterranean diet and physical activity in middle and late adulthood.

Regarding the relationship between physical activity and body shape pressure, a positive relationship is observed for middle and late adulthood, with a negative link for early adulthood.

Finally, this research shows that although adulthood is a period in which the habits acquired in adolescence are reproduced, there are a large number of physical, health and psychological differences in each of its phases, highlighting the importance of psychological, physical and dietary education for future adults.

### Funding
The authors received no funding for this work.

### Competing Interests
Georgian Badicu is an Academic Editor for PeerJ.

### Author Contributions
- Eduardo Melguizo-Ibáñez conceived and designed the experiments, performed the experiments, analyzed the data, prepared figures and/or tables, authored or reviewed drafts of the article, and approved the final draft.
- Gabriel González-Valero conceived and designed the experiments, performed the experiments, analyzed the data, prepared figures and/or tables, authored or reviewed drafts of the article, and approved the final draft.
- Georgian Badicu performed the experiments, prepared figures and/or tables, authored or reviewed drafts of the article, and approved the final draft.
- Wilhelm Robert Grosz analyzed the data, prepared figures and/or tables, and approved the final draft.
- Marius Bazgan analyzed the data, prepared figures and/or tables, and approved the final draft.
- Pilar Puertas Molero conceived and designed the experiments, performed the experiments, prepared figures and/or tables, authored or reviewed drafts of the article, and approved the final draft.

## Human Ethics

The following information was supplied relating to ethical approvals (*i.e.*, approving body and any reference numbers):

The study was conducted according to the guidelines of the Declaration of Helsinki and approved by the Research Ethics Committee of the University of Granada (2966/CEIH/2022).

## Data Availability

The raw data is available in the Supplemental Files.

## Supplemental Information

Supplemental information for this article can be found online at http://dx.doi.org/10.7717/peerj.14652#supplemental-information.

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
