# Peer review of "Mass media pressure on physical build, psychological well-being and physical-healthy profile. An explanatory model in adulthood"

_PeerJ, doi:10.7717/peerj.14652_

## Round 0.1 · original submission · Minor Revisions

Dear authors,

Reply carefully to reviewers' comments and resubmit.

Reviewer 1 ·

Basic reporting

First of all, I would like to thank you for the opportunity to carry out the review of this research.

This is a descriptive study where, through the proposal of a structural equation model, the aim is to study the effect of these in terms of the adult stage in which they find themselves.
The study presents a theoretical framework consisting of basic and current bibliographical references on the problems addressed. Likewise, the methodology shows a good fit and is very well described.

Likewise, the presentation of the results is clear and the discussion responds to the findings obtained. In addition, the conclusions respond to the research objectives and hypotheses.
Although it is a good study, I believe that the discussion should include more current bibliography on the problems addressed. For example: https://revistas.um.es/sportk/article/view/524351
In addition, the list of bibliographical references should be revised, as there are some that do not appear in the text.

Experimental design

This is a descriptive study where, through the proposal of a structural equation model, the aim is to study the effect of these in terms of the adult stage in which they find themselves.
The study presents a theoretical framework consisting of basic and current bibliographical references on the problems addressed. Likewise, the methodology shows a good fit and is very well described.

Validity of the findings

Likewise, the presentation of the results is clear and the discussion responds to the findings obtained. In addition, the conclusions respond to the research objectives and hypotheses.
Although it is a good study, I believe that the discussion should include more current bibliography on the problems addressed. For example: https://revistas.um.es/sportk/article/view/524351
In addition, the list of bibliographical references should be revised, as there are some that do not appear in the text.

Reviewer 2 ·

Basic reporting

No comment.

Experimental design

No comment.

Validity of the findings

No comment.

Additional comments

I would like to thank the journal Peer J as well as the publisher for the opportunity to review this study. I believe it is a necessary study as the research deals with a current issue assessed throughout the different stages of adulthood, in such a way that it helps to understand the issue addressed in a very comprehensive way.

Beginning the review with the introduction, I believe that the theoretical framework is very well-founded, as all the variables are contextualized and the problem to be studied is set out in a very clear way.

In terms of methodology, it is very well structured, contextualizing all the information in different sections that help the reader to understand it. I would like to highlight the importance of the statistical data carried out. In this case, the authors have carried out a reliability analysis of the instruments used. In addition, in order to carry out the ANOVA, they have previously carried out a normality analysis using the Kolmogorov-Smirnov test.

For the ANOVA they studied the effect size following Cohen's criteria. For the models of equations they used the McDonald and Marsh criteria together with the Bentler criteria. Considering the latter criteria, it is true that the two studies where the adjustment indices are studied are fundamental for carrying out this type of analysis, however, I would add a more current quote to contextualize these adjustments in a more current way.

Continuing with the results, these respond in a clear way to the research objectives and hypotheses and are discussed in a very clear way in the discussion. Finally, I consider this research to be publishable, however I believe that it would be necessary to update the references for the study of structural equation modelling.

Annotated reviews are not available for download in order to protect the identity of reviewers who chose to remain anonymous.

Reviewer 3 ·

Basic reporting

no comment

Experimental design

no comment

Validity of the findings

no comment

Additional comments

Congratulations to the authors for their work. The study wants to analyze, identify and establish the relationships between media pressure and physical build on psychological well-being, physical activity and the Mediterranean diet. The study is well conducted with the right scientific rigor; however, I would like to suggest that the authors include graphs showing the results of the questionnaires in relation to age (curved line graphs). On the one hand, this would give more clarity on the results obtained and on the other hand it could give indications for future studies

---

## Round 0.2 · accepted · Accept

Dear authors,

You have diligently followed the suggestions of the reviewers and made the appropriate changes.

The manuscript is now worthy of publication in PeerJ.